# The Role of Art Expertise and Symmetry on Facial Aesthetic Preferences

**Luis Carlos Pereira Monteiro** [1,2], **Victória Elmira Ferreira do Nascimento** [3], **Amanda Carvalho da Silva** [4], **Ana Catarina Miranda** [5], **Givago Silva Souza** [2,4] **and Rachel Coelho Ripardo** [5,*]

1 Neuroscience and Cell Biology Graduate Program, Institute of Biological Sciences, Federal University of Pará, Belém 66075-110, Brazil; luis.monteiro@icb.ufpa.br
2 Center for Tropical Medicine, Federal University of Pará, Belém 66055-240, Brazil; givagosouza@ufpa.br
3 Faculty of Psychology, Institute of Philosophy and Human Sciences, Federal University of Pará, Belém 66075-110, Brazil; victoriaferreirax@gmail.com
4 Faculty of Biological Sciences, Institute of Biological Sciences, Federal University of Pará, Belém 66075-110, Brazil; amandasfurtado@gmail.com
5 Neuroscience and Behavior Graduate Program, Center for Behavioral Theory and Research, Federal University of Pará, Belém 66075-110, Brazil; catarina.miranda@gmail.com
* Correspondence: rcripardo@ufpa.br

**Abstract:** Humans, like other species, have a preference for symmetrical visual stimuli, a preference that is influenced by factors such as age, sex, and artistic training. In particular, artistic training seems to decrease the rejection of asymmetry in abstract stimuli. However, it is not known whether the same trend would be observed in relation to concrete stimuli such as human faces. In this article, we investigated the role of expertise in visual arts, music, and dance, in the perceived beauty and attractiveness of human faces with different asymmetries. With this objective, the beauty and attractiveness of 100 photographs of faces with different degrees of asymmetry were evaluated by 116 participants with different levels of art expertise. Expertise in visual arts and dance was associated with the extent to which facial asymmetry influenced the beauty ratings assigned to the faces. The greater the art expertise in visual arts and dance, the more indifferent to facial asymmetry the participant was to evaluate beauty. The same effect was not found for music and neither for attractiveness ratings. These findings are important to help understand how face aesthetic evaluation is modified by artistic training and the difference between beauty and attractiveness evaluations.

**Keywords:** symmetry; aesthetics; preference; art experts; human faces

## 1. Introduction

Symmetry permeates various elements that can be perceived by humans and can be present in natural or artificial objects [1]. Many researchers have investigated how humans perceive and react to symmetry, especially the symmetry of visual stimuli [2]. This type of symmetry is defined by patterns of repetition through one or more axes in an image [3]. Symmetry based on only one axis, also called bilateral symmetry, is a topic of special interest in the study of visual symmetry, as it is usually more quickly processed and better evaluated than symmetry measures based on more axes [4].

However, symmetry does not have to be just visual or spatial. Examples of symmetry can be found in musical compositions and movements, such as in dance. In music, symmetry is an almost basic condition for its production, as symmetrical patterns are constantly observed in several classical works over the centuries, anchoring melodic processes, rhythm, and harmonic constructions [5,6]. For dance, laterality is an important condition to dictate the direction of choreographic productions. Symmetry, in this case, would be linked to the coordination and use of both sides of the body, an orientation that would even reflect the functional organization of the nervous system [7,8].

The preference for symmetry is evident among several taxa, such as insects [9–11], fishes [12–14], birds [15,16], non-human primates [17,18], and humans [1,2]. In particular, the human preference for symmetry has been widely studied [2]. It is supported by cross-cultural studies but varies according to the stimulus type [19]. Symmetrical patterns tend to be preferred in abstract patterns [20–22], flowers [23], faces [24], and, to a lesser extent, in works of art and landscapes [25,26]. The preference for symmetry is evident not only through explicit evaluations, such as numerical classifications but also through implicit evaluations, such as the relationship between symmetry and positive valence words [27].

The literature presents two non-mutually exclusive explanations for the preference for symmetry in humans: the theory of evolutionary advantage and the theory of perceptual bias (see Treder [1] for a literature review). The theory of evolutionary advantage relates the preference for symmetry to sexual selection, with symmetry being a sign of genetic quality and stability of the physical development of possible partners [28]. This hypothesis is supported by face evaluation studies, which find a consistent positive relationship between attractiveness and symmetry in different cultures [29–31]. However, that relationship seems to change through the different phases of human development [32–34]. The perceptual bias theory, in turn, relates the preference for symmetry as a by-product of the ease of processing symmetrical objects by the visual system [35]. Such objects would require less processing effort, as they present repeated visual information. This processing facility is often called perceptual fluency [36,37].

However, there are conflicting results in other studies. For example, the results of several studies have failed to find the relationship between facial and body symmetry with health, going against what was expected for the evolutionary advantage view [38,39]. On the other hand, some research has found gender differences in the assessment of symmetry in neutral objects [40], abstract patterns [41], and faces [42,43]. Such mixed results indicate that sexual selection and perceptual fluency, alone, are not sufficient to explain the preference for symmetry.

In addition, small degrees of asymmetry are always found in natural organisms considered symmetrical [44]. Therefore, studies manipulating faces and landscapes [25,45] for perfect symmetry show that this is not preferred, as perfect symmetry can be perceived as artificial or unnatural [46]. This preference for symmetry, moreover, can be affected by different factors, such as age [47,48], sex [41], and the observer's expertise in arts [49–51].

Art expertise, which involves theoretical knowledge and amateur or professional experiences in arts, is one of the inter-individual differences that influence aesthetic experience [52]. Several models trying to understand the factors affecting aesthetic appreciation and judgment suggest the importance of art expertise [53–57]. In fact, experienced artists differ from the untrained population in several aspects: in the judgment [58,59], interest [60,61], and emotional response [62] to works of art, as well as in visualization strategies [63,64], in the preference for abstract or representational images [65,66], and the preference for complexity [67,68]. In addition, there is evidence that art expertise modulates neural activity during aesthetic appreciation and artistic creation for the visual arts [69,70], music [71,72], and dance [73,74].

It has been observed that the greater the artistic expertise in visual arts, the more beauty is attributed to objects that are not symmetrical [49–51]. The increase in complexity and abstraction, and, at some level, the symmetry variation in works of art seem to impact lay people more negatively than experienced visual and other artists [75]. The influence of art expertise in the evaluation and aesthetic preference is also observed in other artistic modalities, such as music, in the evaluation of sounds [76,77], and dance, in the evaluation of movement [78,79].

Recent research has found that the preference for symmetry in abstract images is less pronounced in visual artists than in the general population [49–51], but it could be asked if this pattern would also be applied for the preference for symmetry in human faces in visual artists. In addition, it is well established that cross-modal interactions between different sensory modalities can impact individual perception [80,81] and that stimuli modality-

specific (visual/auditory) effects on aesthetic judgments have already been reported [82]. Then, it is reasonable to question how the expertise in non-visually dominant arts such as music and dance would have an impact on the symmetry visual perception and visual aesthetic judgments. So far, it is unknown how the level and type of art expertise could influence the preference for facial symmetry. Thus, the present study aims to assess how the perception of beauty and attractiveness for human faces is influenced by the symmetry of the observed face and the observer's expertise in different types of arts.

## 2. Materials and Methods

### 2.1. Participants

A total of 116 participants recruited online through a university mailing list (registration form in Supplementary file S1) were tested in person. Participants were between 18 and 51 years old (mean ± SD = 24.3 ± 5.27), similarly distributed to biological sex (55 male, 60 female, 1 preferred not to respond), predominantly residents in the capital of the state of Pará, Brazil ($n$ = 95; 81.9%), mostly university students ($n$ = 68; 58.6%), most ($n$ = 82; 70.7%) had an individual monthly income of less than one Brazilian Minimum Wage (R$ 998.00 in 2019) and a family monthly income between 1 and 2 minimum wages ($n$ = 37; 31.9%), had normal or corrected to normal vision, did not use psychotropic drugs, did not have a diagnosis of neuropsychiatric diseases, and most were predominantly heterosexual ($n$ = 66; 56.9%), with the remainder being similarly distributed among bisexuals ($n$ = 26, 22.4%) and predominantly homosexuals ($n$ = 24, 20.7%).

### 2.2. Instruments

#### 2.2.1. Sociodemographic Questionnaire

The sociodemographic questionnaire is in the supplementary materials (Supplementary file S2) and aims to obtain general information about the participants regarding age, sex, marital status, and education, among others. To confirm that the participant was able to carry out the research, the questionnaire also included questions about visual problems and previous diagnoses of neuropsychiatric diseases, as well as about the use of psychotropic drugs.

#### 2.2.2. Arts Expertise Questionnaire

The Arts Expertise Questionnaire is in the supplementary materials (Supplementary file S3) and aims to stipulate the participant's expertise in three artistic modalities: visual arts, music, and dance. It presents items related to formal education, professional experience, and skills in the artistic modalities of interest. This instrument was based on the Art Expertise Questionnaire [70] that include questions for visual arts and music. In the present work, we added adapted questions to also assess dance expertise. The instrument has 45 objective items divided equally into three sections: visual arts, dance, and music (i.e., 15 items for each section). The sum of responses related to visual arts, dance, or music was used as the score of expertise in each art modality.

To ensure the validity of the questionnaire in measuring expertise, we also apply the following validated instruments: (i) Part A (art interest) of the Vienna Art Interest and Art Knowledge Questionnaire (VAIAK) [83] and (ii) Visual arts, Music and Dance sections of the Creative Achievement Questionnaire (CAQ) [84]. The scales mentioned were used since art interest and creativity are variables related to art expertise [59,83]. Each participant was also classified as a layperson ($n$ = 48), visual artist ($n$ = 27), dancer ($n$ = 19), or musician ($n$ = 22) according to their professional area or academic background in the artistic modalities (items 13 and 14 of the sociodemographic questionnaire, respectively). Thus, to be classified as an artist in one of the art modalities (visual arts, music, or dance), the participant needed to be an undergraduate student or a professional in that art modality.

### 2.3. Stimuli

For the present study, we selected 100 photos of faces (50 female and 50 male) with neutral expression from the Chicago Face Database (CFD). The CFD includes 597 high-

resolution photographs of faces of men and women of different ethnicities, aged between 18 and 65 years. The CFD was initially built by Ma et al. [85] and is continuously updated at chicagofaces.org. The facial asymmetry measurement of each face was obtained using geometric morphometric techniques for object symmetry as described in Klingenberg [86]. Object symmetry implies that both sides of the face are mirror images of each other. To run this analysis, the Face++ API (Megvii Technology, faceplusplus.com, accessed on 20 December 2021) was used to automatically add 83 landmarks on selected faces (Figure 1A). Face++ API has been shown to be reliable for obtaining facial landmarks in other works [87,88].

Following this, data from facial landmarks coordinates underwent a Procrustes superimposition—a technique that adjusts the scale, rotation, and translation of shapes without losing their biological variability. These new coordinates data resulting from the Procrustes superimposition were used to calculate facial asymmetry using a mixed two-way ANOVA model. The factors of this ANOVA were the individual and the side of the face from which the landmarks were extracted. As this ANOVA was performed using the Procrustes superimposition data, it is called Procrustes ANOVA. From Procrustes ANOVA it is possible to extract individual facial asymmetry scores, being one measure of absolute asymmetry based on the Procrustes distances and one measure of the relative magnitude of asymmetries based on Mahalanobis distance that is corrected for error variance in the sample. Procrustes superimposition and Procrustes ANOVA were performed in the MorphoJ software [89]. For this study, we used Mahalanobis distances, which avoid multicollinearity problems [90]. The full distribution of the facial asymmetry scores is shown in Figure 1B with examples of faces with the minimum, median, and maximum asymmetry.

### 2.4. Procedure

#### 2.4.1. Instruments Application

At first, the participant responded on a computer in a room at the Federal University of Pará through the Google Forms web application to the (i) Informed Consent Form; (ii) Sociodemographic Questionnaire, and (iii) Arts Expertise Questionnaire with CAQ and VAIAK subscales (Figure 1C). Data collection was carried out in an air-conditioned room at the Behavior Theory and Research Center of the Federal University of Pará. The room had a table, a CPU, and a screen monitor, where the questionnaires were applied and where the face evaluation was carried out. The screen resolution was 1024 × 768 pixels with a 60 Hz refresh rate. Participants were positioned at a fixed distance of 55 cm from the screen during image evaluation. Participants completed the instruments using a standard wired keyboard and computer mouse. The participants respond to the instruments in private, without interference from the researcher responsible, who left the room during the procedure.

#### 2.4.2. Face Evaluation Task

After the application of the Informed Consent Form, Sociodemographic Questionnaire, and Arts Expertise Questionnaire, the face evaluation task was explained to the participant. Subsequently, the participant was asked to briefly explain the task to be performed, in order to ensure understanding. Most participants understood the experiment and, in cases where there were doubts, these were resolved before the experiment was carried out.

The presentation of stimuli and behavioral data collection was prepared using MAT-LAB software version R2018a [91]. In the face evaluation task, 100 photos of human faces (50 female and 50 male) with different levels of symmetry were sequentially presented to each participant in a random order (in order to prevent some photos from being recurrently better evaluated than others by presentation order bias). For each of these photos, the participant evaluated the beauty and attractiveness of the face presented. This task followed the following sequence, repeated for each face (Figure 1C): (i) presentation of a fixation cross for 1 s, (ii) presentation of the face for 3 s, (iii) presentation of the question 'How beautiful do you think is the face you just saw?' along with a rating scale from 1 to 7 (from

not beautiful to extremely beautiful) until the participant's answer; and (iv) presentation of the question 'How attractive do you think is the face you just saw?' along with a rating scale from 1 to 7 (from not attractive to extremely attractive) until the participant's answer. Once the question was answered, it was no longer possible to modify the answer, as the presentation immediately moved to the subsequent screen.

### 2.5. Data Analysis

We used One-way ANOVA with Tukey HSD post hoc tests to compare the scores of laypersons, visual artists, dancers, and musicians on the Arts Expertise Questionnaire in order to verify the validity of this instrument for measuring expertise (see Section 2.2.2 Arts Expertise Questionnaire for an explanation of how these groups were divided). We used Kruskal-Wallis with Dunn's post hoc tests to compare the scores of laypersons, visual artists, dancers, and musicians on Part A of VAIAK and visual arts, music, and dance sections of the CAQ. A non-parametric strategy was used since assumptions of normality and homoscedasticity were not satisfied. Furthermore, associations between art expertise, creativity (CAQ), and art interest (Part A of VAIAK) scores were investigated using Pearson correlation.

We used a one-way repeated measures ANOVA to test whether the average scores of perceived beauty and attractiveness were equivalent. The visual inspection of the model residuals did not show deviations from the assumptions of normality. Through a repeated measures correlation test ($r_{rm}$), we also investigated the existence of a correlation between perceived beauty and attractiveness scores. For both tests, we included the ID of the participants as a random effect.

We used linear mixed-effects models (LMMs) to test whether beauty and attractiveness scores were affected by the facial asymmetry of the observed face. We included random intercepts and slopes within participants and intercepts within stimuli as random effects. Following Clemente et al. [82], we extracted individual slope estimates from the random-effects structure of LMMs as a measure of aesthetic sensitivity—the extent to which a particular sensory feature affects someone's aesthetic valuation [92]—to facial asymmetry for each participant. In this case, negative slope values indicate lower preference for facial asymmetry, which is expected for most participants, whereas positive values indicate greater preference for facial asymmetry. Shapiro–Wilk test was used to assess the distributions' normality of individual slopes. Skewness and kurtosis are shown when the Shapiro–Wilk test *p*-value < 0.05.

In addition, we used multiple linear regression models to test whether individual aesthetic sensitivity to facial asymmetry is affected by the participants' art expertise. In all LMMs and multiple linear regression models, fixed effects were previously centered and scaled (to avoid multicollinearity problems). A parametric modeling strategy was used since (i) visual analysis of the residuals revealed approximately normal distributions and (ii) parametric models have been successful in modeling Likert and other numerical rating scales with more than 5 points (7 points in the present study) and with a sample size larger than 30 [93].

The level of significance was set at α = 0.05 for all statistical analyses. The statistical analyses and the respective visualizations were made using the R software version 4.1.2 [94] and the packages lmertest [95], rmcorr [96], effects [97], and ggplot2 [98].

### 2.6. Ethical Considerations

Before the experiment, participants were informed through the Informed Consent Form about their rights, the topic addressed, and the associated risks and benefits. All methods were performed in accordance with the Declaration of Helsinki, and the protocol was approved by the Ethics Committee of the Center for Tropical Medicine from the Federal University of Pará (report #3.220.201/2019).

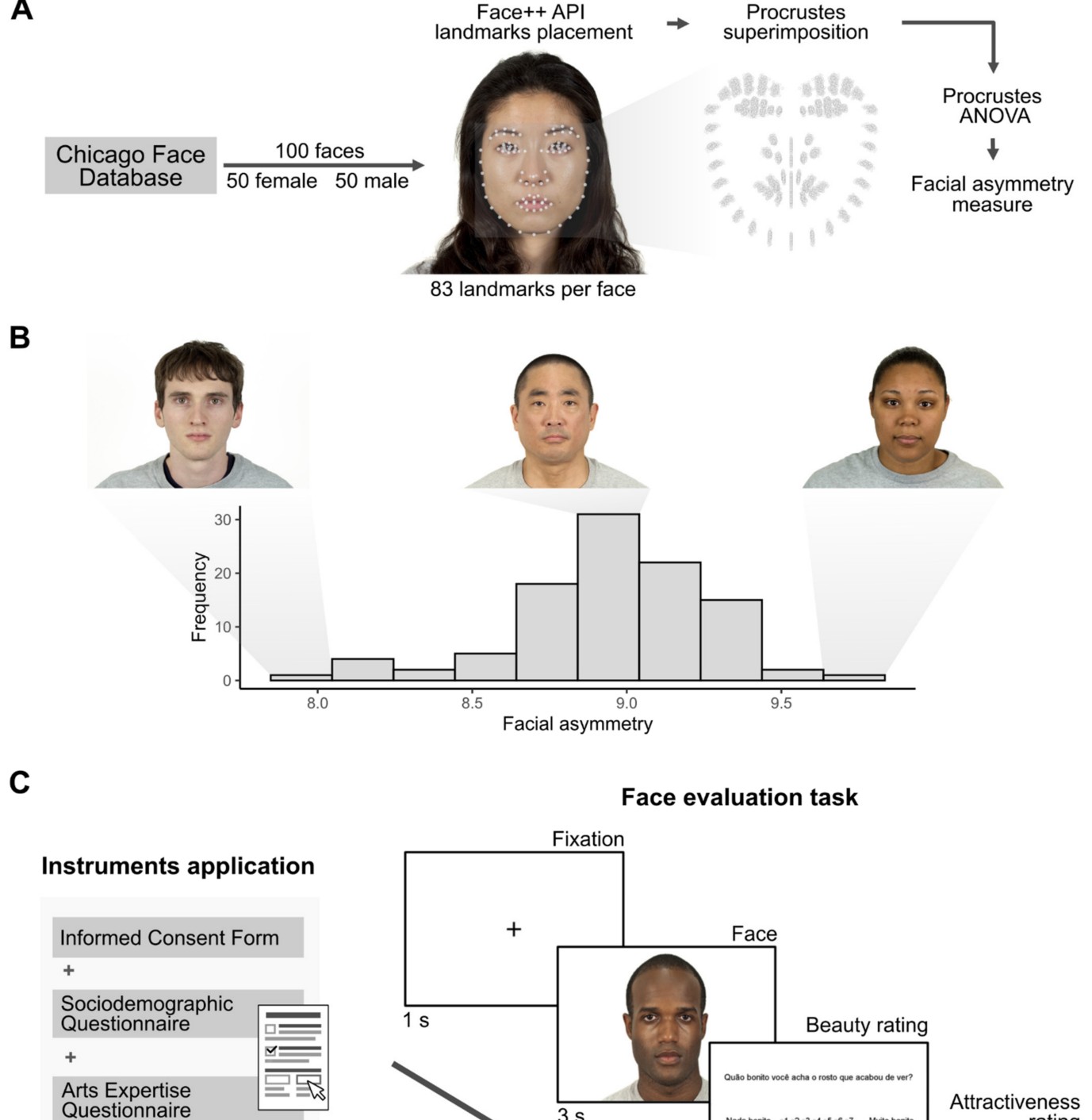

**Figure 1.** Measurement of facial asymmetry on selected faces (**A**), distribution of the facial asymmetry

scores (**B**), and experimental procedure (**C**). In (**A**) 100 faces were randomly selected from the Chicago Face Database (CFD) [85]. Face++ API was used to automatically add 83 landmarks on each selected face. Data from landmarks underwent a Procrustes superimposition and a Procrustes ANOVA analysis, from which individual facial asymmetry values were extracted. In (**B**) the full distribution of the facial asymmetry scores is shown. The photos above the histogram are, from left to right, examples of faces with minimum, median and maximum asymmetry scores. In (**C**) participants responded to the Informed Consent Form, a Sociodemographic Questionnaire, and Arts Expertise Questionnaire. Subsequently, participants performed the face evaluation task. On the first screen, a fixation cross is shown, followed by a face drawn from the CFD, and on the third and fourth screens, respectively, the questions: 'How beautiful do you think is the face you just saw?' and 'How attractive do you think is the face you just saw?' answered on a 1–7 numerical rating scale.

## 3. Results

### 3.1. Differences across Groups in Art Expertise, Creativity, and Art Interest

Although we used art expertise as a continuous variable, we artificially divided the sample into groups of laypersons ($n$ = 48), visual artists ($n$ = 27), dancers ($n$ = 19), and musicians ($n$ = 22) to check if the Arts Expertise Questionnaire was effective in quantifying expertise. Thus, it is expected that the group of experienced artists (students and professionals in the field) obtain higher scores in their respective artistic areas than participants from other groups.

We found an effect of group on visual arts ($F_{(3112)}$ = 36.14, $p < 0.001$), dance ($F_{(3112)}$ = 53.96, $p < 0.001$) and music ($F_{(3112)}$ = 35.48, $p < 0.001$) scores. Visual artists scored higher than laypersons ($p < 0.001$), musicians ($p < 0.001$), and dancers ($p < 0.001$) on the visual arts expertise section. Dancers also scored higher than laypersons in visual arts expertise section ($p = 0.035$). Dancers scored higher than laypersons ($p < 0.001$), visual artists ($p < 0.001$), and musicians ($p < 0.001$) on the dance expertise section. Furthermore, musicians scored higher than laypersons ($p < 0.001$), visual artists ($p < 0.001$), and dancers ($p < 0.001$) on the music expertise section. The distributions of art expertise scores in the different modalities are shown in Figure 2.

We found an effect of group on the creativity (CQA) scores in the visual arts ($\chi^2_{(3)}$ = 53.06, $p < 0.001$), music ($\chi^2_{(3)}$ = 49.35, $p < 0.001$), and dance ($\chi^2_{(3)}$ = 45.18, $p < 0.001$) domains and on art interest (Part A of VAIAK) ($\chi^2_{(3)}$ = 28.16, $p < 0.001$). Visual artists scored higher than laypersons ($p < 0.001$), musicians ($p < 0.001$), and dancers ($p < 0.001$) on the creativity in visual arts domain (Figure 3A). Dancers scored higher than laypersons ($p < 0.001$), visual artists ($p < 0.001$), and musicians ($p < 0.001$) on the creativity in dance domain (Figure 3B). Furthermore, musicians scored higher than laypersons ($p < 0.001$), visual artists ($p < 0.001$), and dancers ($p < 0.001$) on the creativity in music domain (Figure 3C). Laypersons scored less than visual artists ($p = 0.001$), dancers ($p < 0.001$), and musicians ($p < 0.001$) on art interest (Figure 3D). No significant differences for art interest were found between the groups of artists.

The expertise scores of the participants in the three artistic modalities correlated significantly with creativity in their respective domains (visual arts: $r = 0.7$, $p < 0.001$; music: $r = 0.58$, $p < 0.001$; and dance: $r = 0.54$, $p < 0.001$). In addition, art interest correlated significantly with the expertise scores of the participants in all the artistic areas (visual arts: $r = 49$, $p < 0.001$; music: $r = 0.45$, $p < 0.001$; and dance: $r = 0.24$, $p = 0.009$). The relationship between these variables is shown in Figure 3E–H. It is also possible to visualize that visual artists (Figure 3E), musicians (Figure 3F), and dancers (Figure 3G) had higher scores in expertise, art interest, and creativity in their respective domains.

### 3.2. Beauty and Attractiveness Ratings

Regarding the evaluation of the faces by the participants, the beauty ratings (mean $\pm$ SE = 3.45 $\pm$ 0.08) were significantly higher than the attractiveness ratings (mean $\pm$ SE = 2.54 $\pm$ 0.08; $F_{(1,\ 23187)}$ = 2225.5, $p < 0.001$; Figure 4A). Moreover, the beauty and attrac-

tiveness ratings had a significant correlation ($r_{rm}$ = 0.64, 95% CI [0.62, 0.65], $p < 0.001$; Figure 4B).

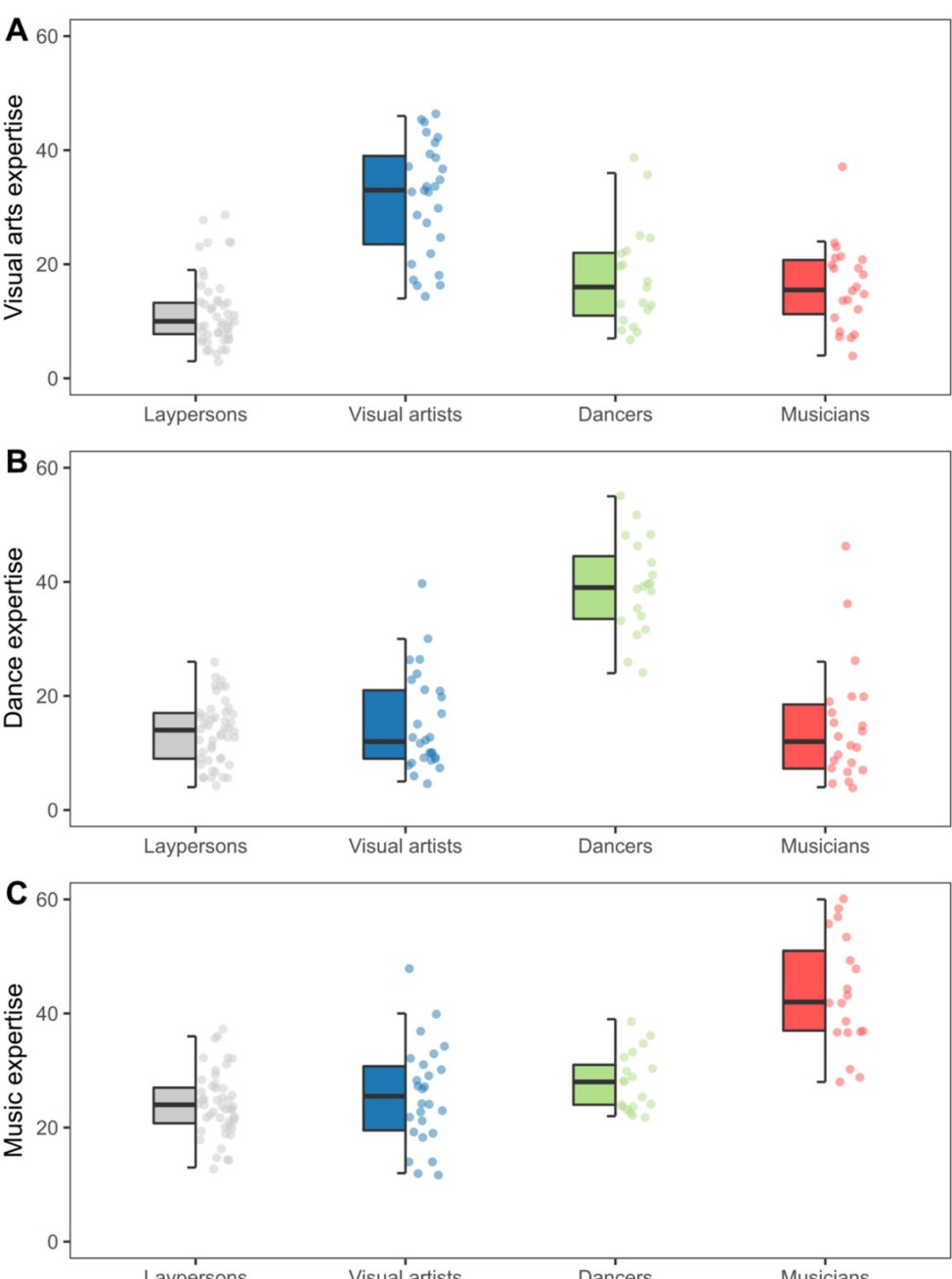

**Figure 2.** Expertise art score in Visual Arts (**A**), Dance (**B**), and Music (**C**) for groups of laypersons and art experts. The sample was divided according to the professional area or academic background of the participants. Horizontal lines indicate the median, the box shows the interquartile range (IQR), and the whiskers are 1.5× IQR. Each dot beside the boxes represents data from a single participant.

*3.3. Influence of Facial Asymmetry on Beauty and Attractiveness Ratings*

Facial asymmetry did not significantly influence beauty ($\beta$ = −0.028, SE = 0.078, $p$ = 0.723) or attractiveness ratings ($\beta$ = −0.038, SE = 0.068, $p$ = 0.578) (Table 1). Individual slopes extracted from the mixed model of beauty rating range from −0.094 to 0.048 (mean ± SE = −0.028 ± 0.029, Figure 5A). Similarly, individual slopes extracted from the attractiveness rating model range from −0.079 to 0.036 (mean ± SE = −0.038 ± 0.026, Figure 5B). For both models, negative slope values indicate lower preference for facial asym-

metry, whereas positive values indicate greater preference for facial asymmetry. Slopes of beauty rating model were normally distributed ($W = 0.99$, $p = 0.945$). Slopes of attractiveness rating model were left skewed distributed ($W = 0.96$, $p = 0.004$, skewness = 0.52, kurtosis = 2.64). Most participants had negative slopes estimates values for both beauty (82.7%, $n = 96$) and attractiveness (89.6%, $n = 104$) rating models, indicating that face asymmetry is poorly rated in both types of evaluation.

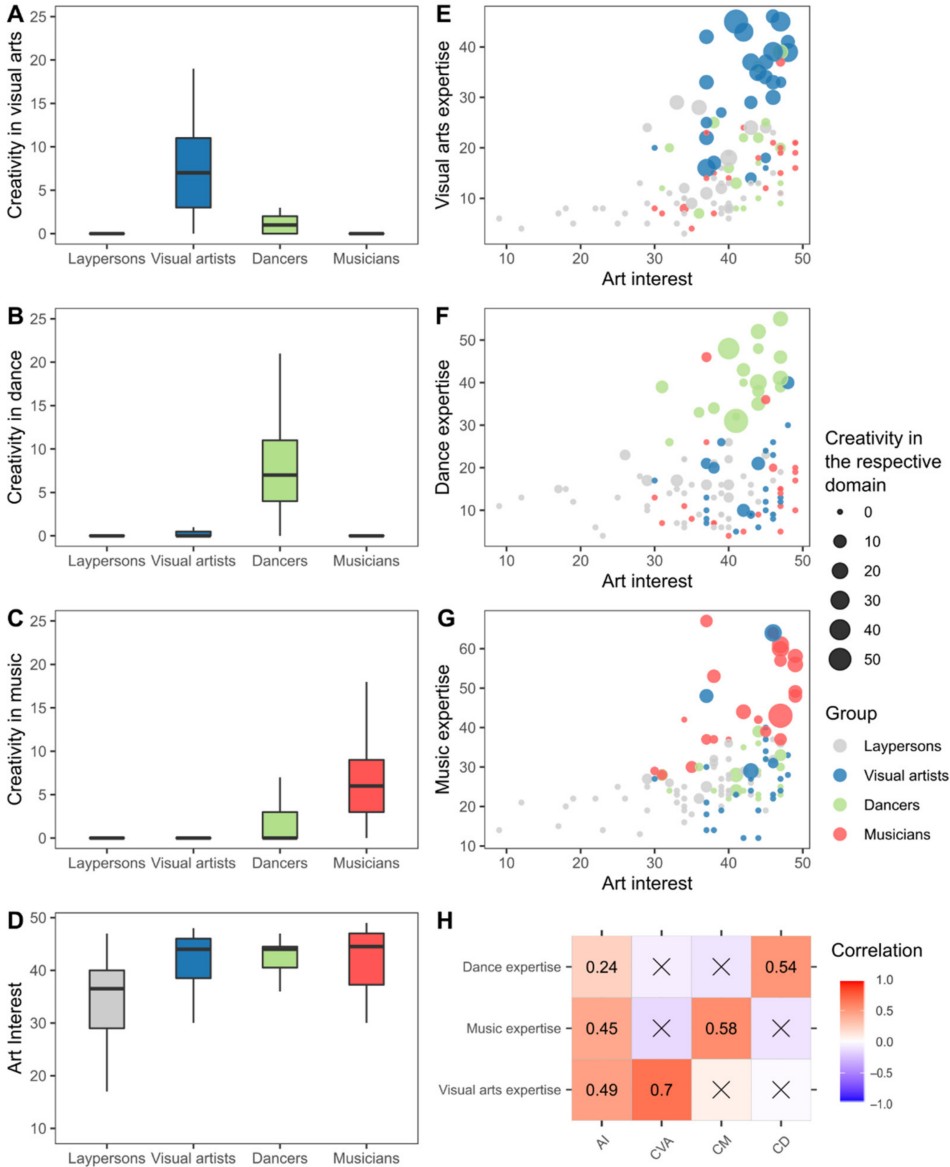

**Figure 3.** Creativity, art interest, and the relationship to art expertise scores in visual arts, dance, and music. In (**A–D**), plots show creativity in visual arts (**A**), dance (**B**), and music (**C**), and art interest (**D**) as a function of group (laypersons, visual artists, dancers, and musicians). Horizontal lines indicate the median, the box shows the interquartile range (IQR), and the whiskers are 1.5× IQR. In (**E–G**), plots show the relationship between art expertise scores, art interest, and creativity. The size of the dots represents the creativity score in the respective domain: visual arts (**E**), dance (**F**), and music (**G**). The groups are color coded. In (**H**), the correlation between art expertise scores and art interest (AI), creativity in visual arts (CVA), dance (CD), and music (CM) domains. The color of tiles is adjusted based on the Pearson correlation coefficient. An X is shown when the correlation coefficient is not significant.

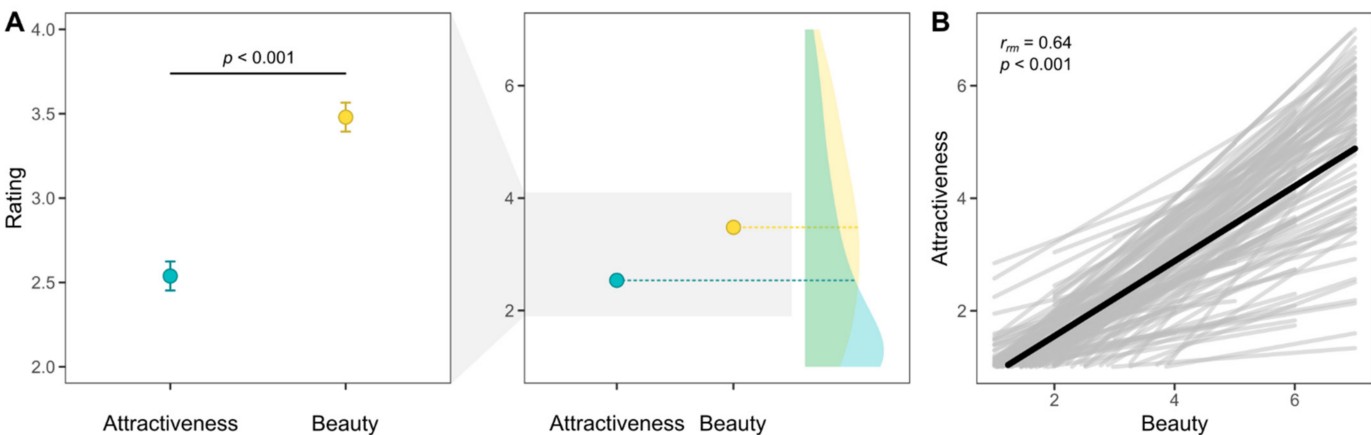

**Figure 4.** Relationship between beauty and attractiveness ratings. In (**A**), first panel shows mean ± SE of beauty and attractiveness ratings (partial effects). The second panel shows the mean attractiveness and beauty ratings (colored dots) and the overlapping kernel density estimate of the full distributions of these ratings. Dashed lines indicate the position of means on their respective distributions. In (**B**) Correlation between beauty and attractiveness (black line) is shown. Gray lines were adjusted separately for each participant using multilevel modeling. In the upper left corner are the correlation coefficient for repeated measures ($r_{rm}$) and the respective *p* value.

**Table 1.** Estimate ($\beta$), standard error (SE) and *p* values for asymmetry in LMMs for beauty rating and attractiveness rating.

|  | Beauty Rating | | | Attractiveness Rating | | |
|---|---|---|---|---|---|---|
|  | $\beta$ | SE | *p* | $\beta$ | SE | *p* |
| Intercept | 3.458 | 0.120 | <0.001 | 2.508 | 0.109 | <0.001 |
| Facial asymmetry | −0.028 | 0.078 | 0.723 | −0.038 | 0.068 | 0.578 |

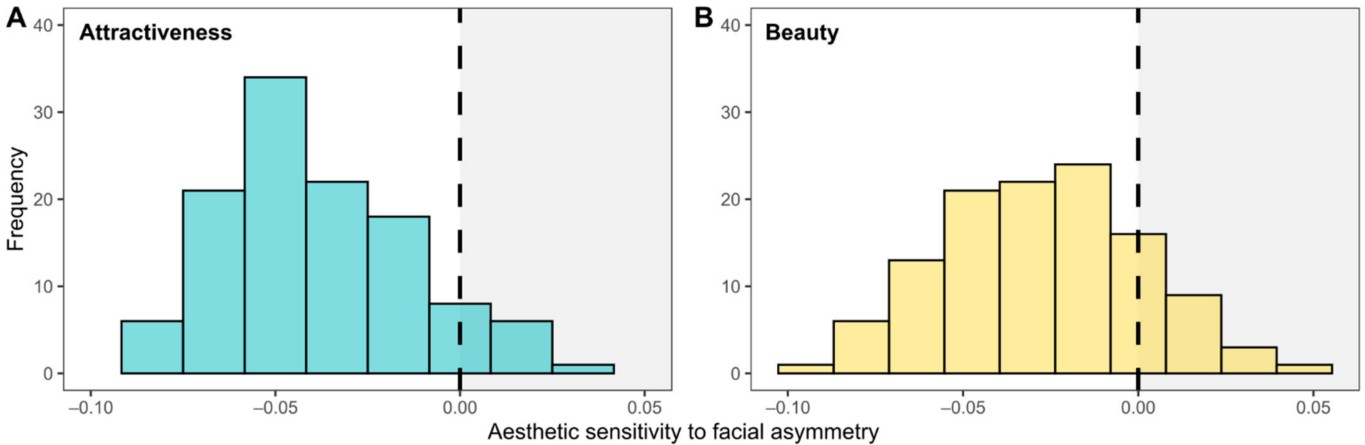

**Figure 5.** Aesthetic sensitivity to facial asymmetry based on individual slopes extracted from the attractiveness (**A**) and beauty (**B**) rating models. Dashed lines indicate zero in *x* axis. Gray area indicates positive individual slope values.

*3.4. Influence of Art Expertise on Aesthetic Sensitivity to Facial Asymmetry*

There was a significant effect of expertise in visual arts ($\beta$ = 0.006, SE = 0.003, *p* = 0.034) and dance ($\beta$ = 0.007, SE = 0.003, *p* = 0.011), but not for music ($\beta$ = −0.0002, SE = 0.003, *p* = 0.565), on aesthetic sensitivity for facial asymmetry in the model based on individual slopes of beauty ratings (Table 2, Figure 6). The variance in aesthetic sensitivity to facial

asymmetry decreased with increasing visual arts and dance expertise (Figure 6). The greater the visual arts and dance expertise, the more predominant were individual slope values close to zero, indicating indifference to facial asymmetry. None of the art expertise was significant in the model based on individual slopes of attractiveness ratings (Table 2). That is, people with higher expertise in visual arts and dance tended to disregard facial asymmetry for beauty ratings, but not for attractiveness ratings. The same trend was not found for music experts.

**Table 2.** Estimate ($\beta$), standard error (SE) and *p* values for Visual Arts, Dance and Music scores in Multiple Linear Regression model for aesthetic sensitivity to facial asymmetry based on individual slopes extracted from the beauty and attractiveness rating models.

|  | Beauty Rating | | | Attractiveness Rating | | |
| --- | --- | --- | --- | --- | --- | --- |
|  | $\beta$ | SE | $p$ | $\beta$ | SE | $p$ |
| Intercept | −0.028 | 0.003 | <0.001 | −0.038 | 0.002 | <0.001 |
| Visual arts | 0.006 | 0.003 | 0.034 | 0.002 | 0.002 | 0.420 |
| Dance | 0.007 | 0.003 | 0.011 | 0.002 | 0.002 | 0.354 |
| Music | −0.002 | 0.003 | 0.565 | 0.002 | 0.002 | 0.537 |

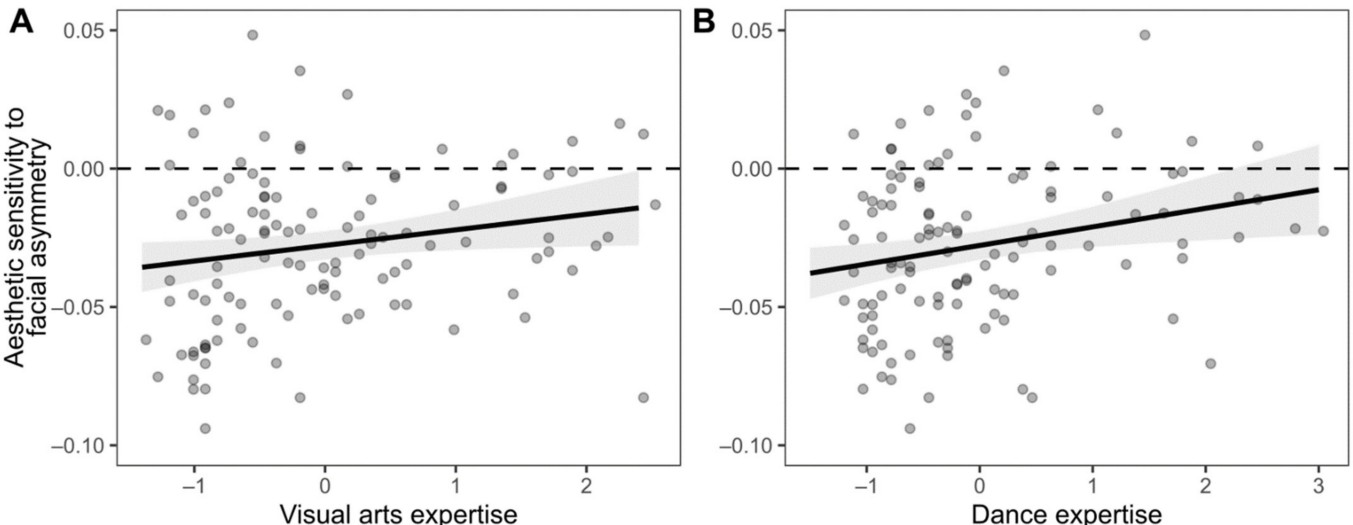

**Figure 6.** Partial effect of Visual Arts (**A**) and Dance (**B**) expertise on aesthetic sensitivity to facial asymmetry based on individual slopes extracted from beauty rating model. Dashed lines indicate zero in *y* axis. Gray area represents the 95% confidence interval.

## 4. Discussion

We investigated for the first time the role of expertise in visual arts, music, and dance, in assessing the beauty and attractiveness of human faces with different asymmetries. Following theoretical models about aesthetic processing, it is expected that art experts and laypersons will differ on their aesthetic evaluation of different sensory features [53–57], including visual symmetry or asymmetry [49–51]. Exploring individual differences, we found that people with higher visual arts and dance expertise tend to disregard facial asymmetry in beauty evaluation of human faces, but not in attractiveness evaluation. The same trend was not found for music experts.

In this work, we use a continuous measure for art expertise as proposed by several authors [70,83,99]. Many studies have used art expertise as a quasi-categorical variable, artificially dividing participants into artists and non-artists—a dichotomy that does not capture the variability within groups concerning this variable [70]. As an alternative, we use a questionnaire that takes into account not only formal education but professional experiences, skills, and other artistic experiences. Art experts scored highest in their specific

areas (e.g., dancers scored higher than other groups in the dance expertise section of our questionnaire). Moreover, art expertise significantly correlated with art interest and with creativity in their specific domains (e.g., visual art expertise scores correlated significantly with creativity scores in visual arts). Such results provide evidence of the instrument's validity to measure expertise.

Most research investigating human preference for faces uses the terms "beauty" and "attractiveness" as synonyms, or simply does not differentiate between them [24,100–102]. In this study, however, we consider beauty and attractiveness as two different variables, as has been done by some authors [103–105]. We observed that although there is a moderate correlation between these two variables, the mean scores for beauty and attractiveness were significantly different. As discussed below, different patterns of individual differences in the assessment of beauty and attractiveness were found, suggesting that they are, in fact, two different variables. In our protocol, after the presentation of the image of the face to be evaluated, the participant first indicated the respective beauty rating, and only afterward the attractiveness rating, so we cannot completely exclude the possibility that the sequence of events may have interfered with the response.

In general, we found that the degree to which facial asymmetry affects beauty evaluation (i.e., aesthetic sensitivity to facial asymmetry) was influenced by participant's visual arts and dance expertise, but not music expertise. Previous research has found an effect of visual arts expertise in the aesthetic evaluation of symmetry/asymmetry in abstract figures [49–51]. The results of Weichselbaum et al. [50] and Gartus et al. [51] indicated that art experts, in general, tended to evaluate stimuli independently from their asymmetry when compared to laypersons. Our results demonstrate that the same trend found for abstract figures can be expected for human faces aesthetic evaluation, not only for visual arts experts but also for dance experts.

While we found a positive effect of the expertise in visual arts and dance on the aesthetic sensitivity to facial asymmetry based on individual differences in the perceived beauty of human faces, we didn't find the same result for music. Visual and auditory stimuli are evaluated differently, and while the appreciation of visual arts and dance relies on vision, the appreciation of music relies on sound. Thus, the differences in the visual assessment of beauty may be related to the peculiarities of each artistic category. These results are consistent with Clemente et al. [82], who found stimuli modality-specific (visual/auditory) effects on evaluative judgments. Moreover, musicians often have a high affinity to symmetrical features, as these are essential to organize the tempo of a melody [5,106].

No evidence for the effect of any of the three areas of art expertise on aesthetic sensitivity to facial asymmetry based on attractiveness ratings was found. This difference between beauty and attractiveness can be explained by the mate choice importance in our species. The mate choice criteria tend to be more stable during human development [107], and therefore should be less influenced by art training. However, it is necessary to take into account the participant's sexual orientation and the gender of the person in the photo evaluated to discuss mate choice accurately. Since our experimental design is not suitable for this type of analysis, we also suggest that further studies take into account these variables.

We also found no effect of facial asymmetry on general beauty or attractiveness ratings. Despite several studies showed that facial asymmetry is an important predictor of facial preference, the magnitude of this effect is relatively small based on meta-analytic estimates [29]. It is also possible that this effect was not found in our study since our sample includes art experts, whose beauty assessment tended to disregard facial asymmetry as commented above.

A possible limitation of this study is the under-representation of highly specialized artists in the areas of interest. This happened in our study because we mostly sought out participants in a general university population, and not in art courses and artistic spaces. Therefore, we suggest that new studies should include more participants with higher art expertise.

The present study concludes that people with different art expertise use asymmetry information differently to evaluate facial beauty. This result can be important in understanding how the facial aesthetic evaluation is modified by this type of training and to give us clues about the way symmetry perception can be affected during human development.

**Supplementary Materials:** The following are available online at https://www.mdpi.com/article/10.3390/sym14020423/s1, Supplementary file S1: Recruitment of research participants form, Supplementary file S2: Sociodemographic questionnaire, Supplementary file S3: Arts Expertise Questionnaire, and Supplementary Data S1.

**Author Contributions:** Conceptualization, L.C.P.M., V.E.F.d.N., A.C.d.S and R.C.R.; Formal Analysis, L.C.P.M., V.E.F.d.N. and A.C.M.; Investigation, L.C.P.M., V.E.F.d.N. and A.C.d.S.; Methodology, L.C.P.M., V.E.F.d.N., A.C.d.S., G.S.S. and R.C.R.; Software, G.S.S.; Supervision, R.C.R.; Visualization, L.C.P.M. and A.C.M.; Writing—original draft, L.C.P.M., V.E.F.d.N. and A.C.d.S.; Writing—review & editing, A.C.M., G.S.S. and R.C.R. All authors have read and agreed to the published version of the manuscript.

**Funding:** This work was supported by research CNPq grant (431748/2016-0). L.C.P.M. received a CAPES scholarship for graduate students. G.S.S. is CNPq Fellows. CNPq Productivity Grant to G.S.S. is 310845/2018-1. CAPES scholarship grant to L.C.P.M. is 88887.497012/2020-00. The funders had no role in study design.

**Institutional Review Board Statement:** The study was conducted in accordance with the Declaration of Helsinki, and approved by the Ethics Committee of the Center for Tropical Medicine from the Federal University of Pará (report #3.220.201/2019).

**Informed Consent Statement:** Informed consent was obtained from all subjects involved in the study.

**Data Availability Statement:** The data presented in this study are available in Supplementary Data S1.

**Conflicts of Interest:** The authors declare no conflict of interest.

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
