# Peer review of "The Role of Art Expertise and Symmetry on Facial Aesthetic Preferences"

_symmetry, doi:10.3390/sym14020423_

Round 1

Reviewer 1 Report

In this paper, the authors had their participants rate the attractiveness and beauty of photos of faces with various degrees of asymmetry. They studied the relationship of art expertise of their participants and the effect of asymmetry in evaluating faces.  In general, this paper is fine though I do have some complaints.

 P4. The method for face stimuli generation is unclear. Procrustes superimposition and Procrustes ANOVA are not well known and thus should be described in better detail. For instance, how the mixed two-way ANOVA was performed? What were the two factors? What was the data analyzed by this ANOVA? Coordinates of the landmarks? Distances between them? Or else? What was the relationship between the ANOVA result and Mahalanobis distance?  Also, it would be nice to show some stimuli of the different asymmetric scores so that the reader could have an idea of the relation between the asymmetry score and the stimuli. Finally, since asymmetry was the major manipulation of this paper, the authors should report the distribution of asymmetry score in their paper. 

Result. The participants made two ratings: attractiveness and beauty. The authors then tried to associate these responses with their artistic expertise. The authors used several ANOVA to analyze their data. However, since attractiveness and beauty were correlated within each participant, the appropriate statistical method should be one MANOVA with both attractiveness and beauty ratings as the dependent variables rather than multiple ANOVAs, one for each rating result. 

Better yet, if the software was available, a structural equation model analysis with expertise and symmetry scores on one side (X) and the two ratings on the other size (Y) would be wonderful. 

Reviewer 2 Report

The current article presents the results of a study that extends the previously reported effects of art expertise on (a-)symmetry preference by showing that expertise in visual art or dance (but not music) is associated with a lesser influence of asymmetry on beauty (but not attractiveness) ratings for faces.

Overall, I think the topic is well-introduced, the study design is sound (albeit the authors also admit not all sequence effects and correlations between ratings were ruled out), and the analyses are appropriate. I want to especially point out that the figures in this article are well-designed and highly informative. Therefore, I can happily recommend this article for publication with minor modifications. I have listed a few issues to consider in chronological order below.

[lines 168-9] What were these procedures used for?

[329] Typo. should be "did not influence"

Figure 5: I highly appreciate that you show the full data as a scatterplot. It does, however, also suggest a slightly more differentiated interpretation of your findings, i.e., that the variance in preference for (a)symmetry is decreasing with increasing expertise and that this decrease is one-sided in that indifference towards (a)symmetry is the more predominant the higher the expertise

[415] Since you did ask participants for their sexual orientation and gender identity: If you want to make strong points regarding attractiveness and mate choice as a driving factor, you will need to show that such effects depend on the observer reporting sexual attraction to the gender of the rated faces.
